# Bayesian Optimization for High-dimensional Urban Mobility Problems

**Seongjin Choi**
Department of CEGE
University of Minnesota
Minneapolis, MN, USA
chois@umn.edu

**Sergio Rodriguez**
Amazon
Seattle, WA, USA
checorh@gmail.com

**Carolina Osorio**
Department of Decision Sciences
HEC Montreal
Montreal, QC, Canada
carolina.osorio@hec.ca

## Abstract

This workshop paper presents a class of important optimization problems that arise in the design of urban mobility digital twins. It presents the open questions in the field and identifies key research opportunities for the communities of Bayesian optimization, uncertainty quantification, and inverse optimization. It open sources the code to tackle an important optimization problem that arises in the design of digital twins: the urban travel demand estimation problem. The code tackles the problem for two road networks: an illustrative toy network and the San Francisco metropolitan network. Code available here: https://github.com/benchoi93/BO_UrbanMobility_Benchmark

## 1 Introduction

Cities worldwide are increasingly developing digital twins of their urban mobility systems. This is due to the increasing complexity of the systems: numerous stakeholders (e.g., travelers, public and private sector mobility service operators, governmental agencies), numerous interacting mobility services (e.g., on-demand ride-sharing, public transportation services), and the shift towards dynamic (e.g., real-time, time-dependent) service operations (e.g., surge pricing for on-demand services, dynamic congestion pricing, real-time speed limits, traffic-responsive traffic signal strategies).

The most high-resolution, or sophisticated, class of such digital twins are known as microscopic traffic simulators. They describe the behavior of both demand and supply in detail. On the demand side, individual vehicles have their own technology (e.g., electric, autonomous, connected), and individual travelers have their own behavior (e.g., aggressive drivers, business travelers with high value of time, different willingness to shift to new travel modes). On the supply side, the operations of the city's infrastructure (e.g., traffic signal plans, congestion pricing policies, fleet of public transportation vehicles) and of the available mobility services (e.g., taxi offerings, bike-sharing, on-demand ride-hailing) are also modeled in detail. The underlying demand models that govern the choices of travelers such as mode choice, departure time choice, and route choice, are most often based on well-established probabilistic models (e.g., random utility models). Hence, the realization of the trip of an individual involves sampling from a number of probability distributions. The resulting simulation of the urban mobility system then involves simulating the trips of a large (e.g., tens or hundreds of thousands) population of vehicles and travelers.

This results in digital twins that yield stochastic costly to compute and nonlinear network performance metrics. However, these models are most often used in a small-sample context, where few (sequential) function evaluations of the simulator are allowed. This is due to both the compute cost and the time constraints that cities, and numerous mobility operators, work with. Hence, there is potential for the research communities of sample-efficient learning, optimization, and more specifically Bayesian

Workshop on Bayesian Decision-making and Uncertainty, 38th Conference on Neural Information Processing Systems (NeurIPS 2024).

optimization (BO) and the uncertainty quantification (UQ), to contribute to advancing the science and the practice of urban mobility planning and operations.

This article discusses the, arguably, most important and difficult open optimization problem that arises when building a digital twin of an urban mobility network. It shares starter code to tackle the problem applied to two road network instances: one for a toy road network, and one for a challenging San Francisco metropolitan area network. The workshop talk that accompanies this paper will provide: (i) a broader discussion of the open problems in the field with concrete ideas on how the BO/UQ communities can contribute, (ii) a more detailed analysis of the most interesting mathematical aspects of the considered class of optimization problems.

## 2   Problem Formulation

The numerous demand and supply models mentioned above, are most often parametric models. Calibration problems that aim to learn or optimize the values of the input parameters are critical to designing a digital twin that is capable of replicating historical traffic patterns. The most critical calibration problem is known as demand calibration, or origin-destination (OD) travel demand calibration. In it's most traditional form, it consists of estimating the expected number of trips (known as the travel demand) that originate from a given urban zone and terminate at another urban zone.

Traditionally, a metropolitan area is partitioned into traffic analysis zones (TAZes). A subset of all pairs of zones is considered, and the corresponding travel demand is estimated for these pairs. Due to this spatial partitioning into discrete zones, the number of considered pairs is often in the tens or hundreds of thousands. Hence, the demand calibration problem can be formulated as a: stochastic (simulation-based) continuous high-dimensional problem with a costly to evaluate loss function that is to be solved with a small simulated sample.

The problem can be formulated as follows,

$$\min_{x \in \Omega} \sum_{i \in \mathcal{I}} \left( y_i^{\mathrm{GT}} - E[Y_i(x; p)] \right)^2, \tag{1}$$

where $y_i^{\mathrm{GT}}$ denotes a statistic obtained from ground truth (i.e., field) traffic data, such as mean vehicular count, speed, travel time, on a given road $i$, and $E[Y_i(x; p)]$ denotes the simulated counterpart derived from the digital twin. The latter depends on the vector of travel demands $x$, a vector of additional parameters $p$, the set of roads in the network where traffic data or measurements is available is denoted $\mathcal{I}$. In it's simplest form, the feasible region $\Omega$ consists of simple upper and lower bound constraints.

This formulation aims to minimize the squared distance between the historical traffic statistics and the corresponding simulated counterparts. The inverse optimization community will recognize Problem (1) as an inverse optimization problem. The challenges of tackling this problem are: (i) $x$ is high-dimensional (in the tens or hundreds of thousands), (ii) $E[Y_i(x; p)]$ is an unknown costly to compute function that can only be estimated via stochastic simulation.

In the workshop talk, we will expand on the following topics. We will present various other formulations, including distribution matching formulations where the goal is to estimate the distribution of $x$ rather than a point estimate. We will also discuss the scientific and practical implications of this being an underdetermined or ill-posed problem. This underdetermination is, in part, due to the fact that traffic measurement sites are sparsely located throughout the network (so the cardinality of $\mathcal{I}$ is small relative to the number of roads in the network). For metropolitan networks, there is a high level of underdetermination. Hence, the importance of both quantifying the level of underdetermination and developing UQ techniques. Such techniques are critical to develop counterfactually robust digital twins that enable stakeholders to evaluate the impact of novel urban mobility policies.

## 3   Case studies

We use the open-source traffic simulation software SUMO (1). The toy network used is known as Quick Start (2) and is part of a set of small instances provided by the SUMO Tutorials. The metropolitan network is the San Francisco (SF) metropolitan area network derived from (3), we consider demand scenarios that differ from those of the original dataset. Summary information of each network is given in Table 1. The first row of the table indicates the total number of modeled

Table 1: Case study summary

| Network | Toy | San Francisco |
|---|---|---|
| # of roads | 26 (14) | 196,421 (197) |
| # of intersections | 2 | 8,215 |
| Dimension of $x$ | 4 | 10 |

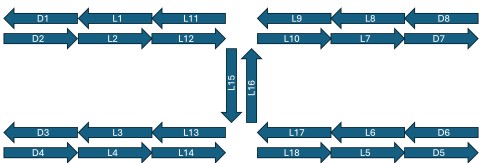

Figure 1: Toy network: network topology

roads as well as, in parenthesis, the number of roads used in the demand scenarios considered in this paper. In particular, for the SF network, we start with a relatively small, yet still non-trivial, 10-dimensional instance that uses 197 roads throughout the network.

Figure 1 depicts the network topology of the toy network. The considered demand instance has two origin locations: roads $D2$ and $D4$. Both origins share the same possible destinations: roads $D7$ and $D5$. This leads to a 4-dimensional instance. Figure 2 depicts the topology of the SF network. For both networks, the ground truth (GT) traffic statistics ($y_i^{\text{GT}}$ of Problem (1)) are based on synthetic data: we assume a given GT travel demand (known as the GT OD) and generate, via simulation, the corresponding traffic statistics. Thereafter, we assume the GT OD is unknown.

We apply a basic Bayesian optimization method with a Gaussian process surrogate. Specifically, we utilize a Matern kernel with $\nu = 2.5$ and Automatic Relevance Determination (ARD) over the input dimensions, where the lengthscale is constrained within the interval $[0.005, 4.0]$. For each network, we run 10 BO restarts. Each restart considers a different random initial sample, which is of size 10 (resp. 30) for the toy (resp. SF) network. The initial sample is based on Sobol sequences within the bounded search space, ensuring a more uniform and low-discrepancy sampling distribution. We employ the q-Log Expected Improvement (qLogEI) as the acquisition function.

Example plots that illustrate the type of analysis that will be presented at the workshop follow. The plots of Figure 3 are convergence plots that depict the, commonly used, normalized root mean square error (NRMSE), as defined below, of the best point $x$ versus the BO epoch.

$$\text{NRMSE}(x) = \frac{\sqrt{\frac{1}{n} \sum_{i \in \mathcal{I}} \left( y_i^{\text{GT}} - y_i^{\text{sim}}(x) \right)^2}}{\frac{1}{n} \sum_{i \in \mathcal{I}} y_i^{\text{GT}}}, \tag{2}$$

where $y_i^{\text{sim}}(x)$ represents the simulated estimate of $E[Y_i(x; p)]$ of Problem (1), and $n$ is the cardinality of $\mathcal{I}$. The solid line is the mean NRMSE across the BO restarts, with error bars that have a half-width

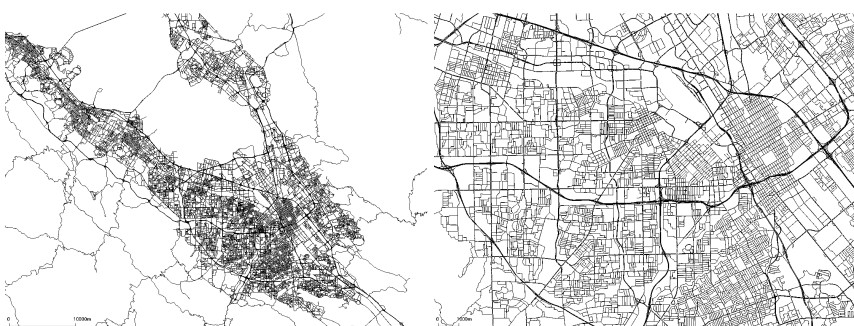

Figure 2: San Francisco: full network topology (left plot), zoomed-in area of SF network (right plot)

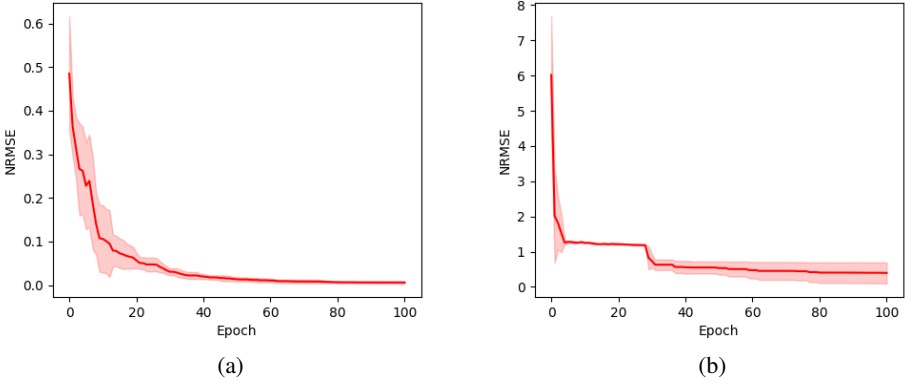

Figure 3: BO convergence: (a) toy network and (b) SF network

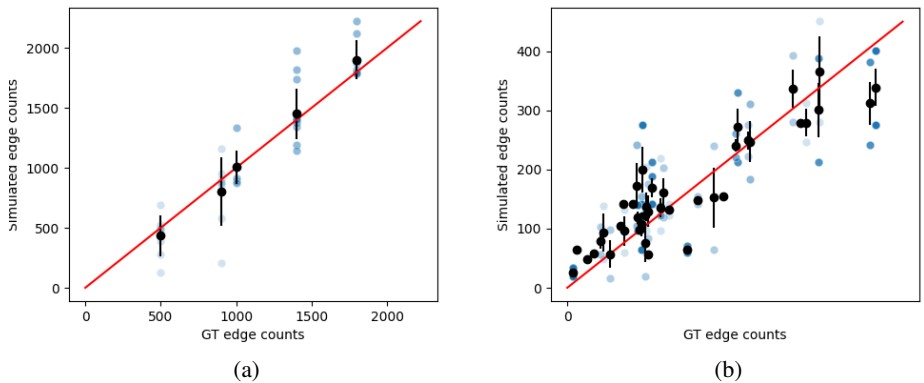

Figure 4: Fit to ground truth data: (a) toy network and (b) SF network.

of one standard deviation. For the type of traffic data considered here, which is vehicular count data, an NRMSE of the order of 0.2 or below is considered an adequate fit of the simulator to the field data.

The plots of Figure 4 are scatter plots that illustrate the fit of the simulated data (i.e., simulated vehicular counts) to the GT field data: the x-axis (resp. y-axis) depicts the GT traffic data ($y_i^{\mathrm{GT}}$ of Problem (1)) (resp. $y_i^{\mathrm{sim}}(\mathrm{x})$ of Eq. (2)). The simulated counts of each restart are depicted as blue circles. The mean simulated count, across restarts, is depicted as a black circle, along with error bars that have a half-width of one standard deviation. The closer the points are to the red diagonal line, the better the fit of the BO-identified travel demand input. As will be illustrated in the workshop, for both networks, BO yields a significant improvement in the fit to the traffic data relative to the initial points.

The workshop discusses opportunities to improve the ability of BO to scale to the high-dimensional instances that are typically tackled by practitioners. In particular, we discuss various kernel specifications that embed simple traffic-physics information and enhance the sample efficiency of BO for this class of problems.

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
