# OpenReview forum: "Bayesian Optimization for High-dimensional Urban Mobility Problems"
_NeurIPS.cc/2024/Workshop/BDU — NeurIPS BDU Workshop 2024 Poster_

### Official Review · Reviewer_T9A8 · 2024-09-23
**Review comment for Submission 73**

**Rating:** 5
**Confidence:** 4

**Review:**

1. The challenges and primary motivation are not clearly articulated. The practical implications of this research need to be discussed in more detail. How does this work translate into real-world applications?
2. The features utilized in the two datasets are not clearly explained. Please provide more details.
3. Please elaborate on the discrepancies between the SUMO-simulated data and the real-world data. How do you address these differences? Additionally, more discussion needs to be included about the issue of time complexity and scalability.

---

### Official Review · Reviewer_HLSY · 2024-09-27

**Rating:** 7
**Confidence:** 3

**Review:**

The paper, "Bayesian Optimization for High-dimensional Urban Mobility Problems", presents an innovative approach to optimizing urban mobility digital twins using Bayesian optimization (BO). It focuses on travel demand calibration in complex networks, such as the San Francisco metropolitan area.

Pros:
- The paper effectively applies BO to realistic, high-dimensional networks, as demonstrated in its case studies on the San Francisco network, providing a clear path for practical implementation
- It highlights the potential of BO for handling costly, simulation-based optimization problems, showcasing its ability to achieve sample-efficient learning in complex urban mobility systems​
- The inclusion of code for travel demand estimation ensures the research is reproducible

Cons:
- The paper relies heavily on synthetic data for validation, which may limit the applicability of its findings to real-world scenarios​

---

### Decision · Program_Chairs · 2024-10-09

Accept (Poster)